# Src Family Tyrosine Kinases in Intestinal Homeostasis, Regeneration and Tumorigenesis

**DOI:** 10.3390/cancers12082014

**Published:** 2020-07-23

**Authors:** Audrey Sirvent, Rudy Mevizou, Dana Naim, Marie Lafitte, Serge Roche

**Affiliations:** CRBM, CNRS, University of Montpellier, Equipe labellisée Ligue Contre le Cancer, F-34000 Montpellier, France; audrey.sirvent@crbm.cnrs.fr (A.S.); rudy.mevizou@crbm.cnrs.fr (R.M.); dana.naim@crbm.cnrs.fr (D.N.); marie.lafitte@crbm.cnrs.fr (M.L.)

**Keywords:** Src, intestinal epithelium, cell signalling, tyrosine kinase, colon cancer, cancer therapy

## Abstract

Src, originally identified as an oncogene, is a membrane-anchored tyrosine kinase and the Src family kinase (SFK) prototype. SFKs regulate the signalling induced by a wide range of cell surface receptors leading to epithelial cell growth and adhesion. In the intestine, the SFK members Src, Fyn and Yes regulate epithelial cell proliferation and migration during tissue regeneration and transformation, thus implicating conserved and specific functions. In patients with colon cancer, SFK activity is a marker of poor clinical prognosis and a potent driver of metastasis formation. These tumorigenic activities are linked to SFK capacity to promote the dissemination and tumour-initiating capacities of epithelial tumour cells. However, it is unclear how SFKs promote colon tumour formation and metastatic progression because SFK-encoding genes are unfrequently mutated in human cancer. Here, we review recent findings on SFK signalling during intestinal homeostasis, regeneration and tumorigenesis. We also describe the key nongenetic mechanisms underlying SFK tumour activities in colorectal cancer, and discuss how these mechanisms could be exploited in therapeutic strategies to target SFK signalling in metastatic colon cancer.

## 1. Colorectal Cancer

Colorectal cancer (CRC) is a heterogeneous disease influenced by specific genetic, epigenetic, and environmental factors [1]. It is one of the leading causes of malignancy-related death worldwide because of its recurrent and invasive nature [2]. The current clinical management of localized tumours involves their surgical removal and adjuvant chemotherapy; however, tumour recurrence and metastatic spread are observed in about 25–40% of patients, resulting in poor prognosis with a 5-year survival rate of 10% [2]. CRC aggressiveness is associated with the epithelial tumour cell’s capacity to promote dissemination and tumour reactivation, through mechanisms that have not been fully elucidated yet [3]. Comprehensive genomic analyses of CRC heterogeneity allowed the classifying of tumours into different molecular subtypes: ultramutated, microsatellite instability-high/hypermutated (MSI), and microsatellite stable (MSS) [4]. Moreover, early-stage CRCs can be grouped into four consensus molecular subtypes (CMS) in function of the driving gene alteration: MSI+ tumours with strong immune infiltration (CMS1), Wnt/beta-catenin proliferative tumours (CMS2) and KRAS mutated and metabolic-deregulated tumours with strong immune exclusion (CMS3), and “mesenchymal” tumours with stromal and innate immune infiltration (CMS4) [5,6]. More recently, sequencing of metastatic CRC samples identified few genetic differences between primary tumours and metastases, but some alterations in the p53 pathway. Specifically, Wnt/beta-catenin signalling alterations were detected in most metastatic samples, suggesting that this oncogenic pathway has an obligate role in metastatic progression [7]. These findings are consistent with the model of CRC progression established using genetically modified mouse models in which mutations in the *Apc* tumour suppressor gene are combined with alterations in Mitogen Activated Protein Kinase (MAPK) signalling, p53, and Transforming Growth Factor-β (TGF-β) signalling [8,9]. This sequencing study also identified important genetic differences between primary CRC localized in the right and left colon [7]. Notably, right-side CRCs were associated with shorter survival, older age at diagnosis, increased numbers of mutations, and enrichment of oncogenic alterations in Phosphatidyl Inositol 3 Kinase (PI3K) and MAPK signalling. Conversely, left-sided tumours relied on nongenetic deregulation of tyrosine kinase (TK) signalling and environmental changes, such as intestinal microbiota, suggesting an important tumour-promoting role for this signalling mechanism in CRC.

## 2. Tyrosine Kinases in CRC

TKs use protein phosphorylation on tyrosine residues as an intracellular signalling mechanism to coordinate epithelial cell communication and fate decision. Their deregulation can lead to carcinogenesis [10]. Indeed, excessive TK signalling has been observed in CRC, including the one from the Receptor Tyrosine Kinases (RTKs) Epidermal Growth Factor Receptor (EGFR) and Vascular Endothelial Growth Factor Receptor (VEGFR) [6]. Genomic studies indicate that these receptors are preferentially activated in CMS2 tumours, although the deregulation of TK signalling and their effectors (e.g., MAPK and PI3K) is observed in all CRC subtypes [6]. Therefore, several TK signalling inhibitors have been developed, and EGFR and VEGFR inhibitors have been approved for the treatment of metastatic cancer. These therapies have been tested also in patients with CRC, but they displayed moderate effects and patient survival was prolonged only for a few months [11]. Although oncogenic SFK TKs are potent drivers of CRC metastasis, clinical trials using small Src-like inhibitors (Srci) in CRC failed [12]. One reason is that it is still unclear how aberrant TK activities contribute to CRC formation because TK-encoding genes are not frequently mutated in this cancer [4]. Therefore, unravelling the underlying mechanisms may lead to the identification of key nongenetic mechanisms by which TKs promote CRC formation, and ultimately to the development of efficient therapies based on effective TK signalling inhibition and patient selection. Here, we review recent findings on Src Family Kinase (SFK) function in intestinal homeostasis, regeneration, and tumorigenesis, and describe essential nongenetic mechanisms underlying SFK tumour activities. We also discuss how these tumour-promoting mechanisms could be exploited in therapies targeting SFK signalling in CRC.

## 3. SFKs in Intestinal Tumours

### 3.1. SFKs

Src, originally identified as an oncogene, is a membrane-anchored cytoplasmic TK that mediates signalling induced by a wide range of cell surface receptors [13,14]. Notably, Src is a master regulator of cell growth and migration induced by extracellular cues. Src is also the prototype of SFK that includes eight members (Src, Fyn, Yes, Lck, Fgr, Hck, Blk and Lyn) of which three (Src, Fyn and Yes) are widely expressed [13,14]. Src shares with the other SFKs a common modular structure formed by the membrane-anchoring SH4 region through lipid attachment (i.e., myristoyl), followed by an intrinsically disordered region named the unique domain (UD), and the SH3, SH2 and kinase domain [13,14]. SH4 contains also a palmitoylation site for membrane anchoring, except in Src and Blk. The kinase domain is bordered by two short regulatory sequences, named the linker and the tail, involved in the tight regulation of the kinase activity to prevent aberrant protein tyrosine phosphorylation. Crystallography studies revealed that intramolecular interactions are part of the mechanisms that control SFK catalytic activity [15]. Notably, phosphorylation of a conserved tyrosine in the tail (Tyr 530 in the human Src sequence) promotes SH2 intramolecular interactions that, combined with the SH3-linker interaction, stabilise the enzyme in an autoinhibited conformation. This Src phospho-regulatory mechanism is conserved in all SFKs and is mediated by the cytoplasmic TKs C-terminal Src Kinase (CSK) and CSK homologous kinase [16]. Disruption of any of the SH2 or SH3-mediated protein interactions or tyrosine phosphatase activity leads to the kinase active conformation. Catalytic de-repression enables SFK autophosphorylation of the activation loop (Tyr419 in the human Src sequence) that further supports the kinase active state [15]. In agreement with this model, stabilization of the enzyme in a de-repressed conformation by somatic mutation or protein association results in constitutive SFK activity that can lead to oncogenic properties [17].

Recent findings revealed additional unsuspected Src regulatory mechanisms involving its UD. This unstructured region of about 70–80 amino acids is conserved in vertebrates and is unique among the different SFKs (e.g., different sequences). Although Src UD function remained mysterious until recently, NMR analyses revealed that this region has a compact, yet highly dynamic structure, described as an intramolecular fuzzy complex [18]. NMR-guided mutations that affect UD-SH3 interactions revealed an essential role for this fuzzy complex in Src signalling leading to CRC cell migration [19]. Moreover, Src can dimerise through involvement of the UD in the binding to a hydrophobic pocket within the kinase domain of the dimeric partner [20]. A biophysical study showed that the Src SH4 domain has dimerisation capacity on its own, suggesting a complex mechanism underlying Src dimerisation [21]. Importantly, Src dimerisation may define a novel regulatory mechanism because it substantially enhances Src autophosphorylation and phosphorylation of selected substrates [20]. Interestingly, SFKs share well-conserved sequence features involving aromatic residues in their UDs, suggesting a similar UD-dependent regulatory region in the other SFKs [18]. 

Finally, emerging evidence supports the existence of an additional regulatory mechanism through Src myristoylation, as described for the cytoplasmic TK Abelson (ABL) where a myristoyl binding pocket in the kinase domain maintains ABL in an inactive state [22]. Structural analyses suggested, but did not confirm yet, the presence of a similar binding pocket in Src [23]. Nevertheless, Moasser et al. reported that Src dimerisation also involves the interaction of the myristoylated N-terminal region with the kinase domain pocket in trans [20]. Surprisingly, Pons et al. discovered an additional myristoyl binding site in Src-SH3, that contributes to Src membrane anchoring [24]. Interestingly this interaction is modulated by the fuzzy complex contained in the UD, suggesting a mechanism linking Src activation and membrane anchoring. Therefore, we predict an important role for the SH4 and the UD in controlling the Src topology at the membrane or the local microenvironment for substrate selection and signalling. Whether this myristoylation-switch mechanism is conserved in other SFKs is currently unknown. Overall, these recent findings uncover a much higher complexity of SFK regulation than previously expected, which may have important implications on the SFK’s oncogenic functions.

### 3.2. SFKs in Intestinal Homeostasis and Regeneration

Genetic analyses in animals established essential physiological roles for SFKs [25]. For instance, constitutive SFK gene knock-out experiments in mice revealed an important function for Src in early development, at least partially shared with Fyn and Yes. Specifically, Src-deficient mice die early after birth because of defects in bones where Src is normally highly expressed. Conversely, combined *Src*, *Fyn* and *Yes* gene inactivation leads to mouse embryonic lethality [25]. Consistent with this, disruption of the SFK negative regulator *Csk* leads to embryonic lethality with excessive tissue SFK activity, indicating that SFK regulation is essential for mouse development [25]. However, other genetic analyses revealed that *Src*, but not *Fyn*, is partly epistatic to the *Csk* gene, consistent with SFK partially redundant functions during development [25]. Then, tissue-specific gene manipulation studies showed important roles for SFKs in epithelial tissues. For instance, Cordero et al. performed SFK gain and loss of function experiments in mouse and fruit fly intestines to address SFK’s physiological role in the intestinal epithelium [26]. *Drosophila* is a useful genetic model to study intestinal homeostasis because, as in mammals, adult fly midgut epithelium is renewed by intestinal stem cells (ISC) [27]. Src42A and Src64B are the two Src-like kinases expressed in *Drosophila*, and are the likely orthologues of Src and Fyn, respectively. However, only Src42A loss of function inhibits ISC proliferation in conditions of homeostasis and stress response to bacterial infection, suggesting a specific Src function in the intestine. Nevertheless, genetic overactivation of any of these SFKs is sufficient to drive ISC hyperproliferation, indicating potential SFK redundant functions above a certain threshold [26,28]. An Src key role was confirmed in a mouse model where *Src* ablation in the intestine prevented ISC proliferation and crypt regeneration after induction of DNA damage by gamma irradiation [26]. However, *Src* ablation alone was not sufficient to affect intestinal homeostasis because of its overlapping functions with Fyn and Yes. In agreement, combined inactivation of *Src*, *Fyn* and *Yes* in the intestine leads to intestinal epithelial cell (IEC) apoptosis and reduction of the number of Paneth cells in the small intestine [26], Conversely, ablation of their negative regulator *Csk* increases IEC proliferative activity and turnover [29] (Figure 1).

Mechanistically, Src drives ISC proliferation through upregulation of EGFR, activation of Ras/MAPK and signal transducer and activator of transcription 3 (Stat3) signalling [26]. This Src function was revealed after intestinal injury induced by irradiation. The mechanism underlying Src-mediated ISC proliferation is not fully clear, but may implicate important intestinal regulators, such as Wnt/beta-catenin signalling [26], that controls proliferation and differentiation of crypt-localised ISCs, and Notch that controls the enterocyte lineage [30] (Figure 1A,B). Src also uses the transcription factor Yes Associated Protein (YAP), an essential sensor of the cell microenvironment structural and mechanical features [31], to mediate epithelial regeneration during intestinal inflammation. This new Src signalling activity was uncovered from a genetic mouse model with persistent intestinal inflammation upon IEC overexpression of gp130, the coreceptor for interleukins of the IL-6 family [32]. These animals display aberrant IEC proliferation and differentiation and are resistant to mucosal erosion. This gp130 activity is mediated by interaction with Src and Yes to phosphorylate the transcription factor YAP on specific tyrosine residues and to induce its stabilisation and nuclear translocation [33]. Surprisingly, this inflammatory mechanism is independent from the effector Stat3 [32]. A similar Src/YAP signalling has been described during intestinal regeneration mediated by dietary and metabolic factors, [34] strengthening the conserved role of Src/YAP signalling in intestinal repair. Src may also induce ISC proliferation through a cell nonautonomous mechanism. Indeed, in *Drosophila* gut, upon bacterial infection, Src activation in enterocytes induces IL-6 expression that leads to ISC proliferation [35]. Similarly, in the mouse, in physiological conditions, intestinal tissue is renewed by Leucine-rich repeat-containing G-protein coupled receptor 5-positive (Lgr5+) ISCs localised at the bottom of the crypt [36]. However, upon severe damage, intestinal epithelium can be regenerated by a distinct mechanism [37] that involves IEC dedifferentiation via transient foetal-like features [38]. Importantly, this regenerative mechanism is mediated by extracellular matrix remodelling that enables biomechanic Src/YAP signalling for efficient tissue repair [38] (Figure 1B). A similar mechanism was reported for Class A basic helix-loop-helix protein 15-positive (Bhlha15+) intestinal secretory precursors that transiently convert into enterocyte progenitors after doxorubicin-induced epithelial injury [30]. In the mouse, elevated SFK activity induced by *Csk* ablation in IECs activates an additional Rac signalling mechanism to promote IEC proliferation [29]. Interestingly, these animals display enhanced susceptibility to colitis induced by the chemical irritant dextran sodium sulphate (DSS), due to low epithelial barrier function caused by tight junctions reduction [39].

Finally, SFKs have also specific functions in immune cells. Indeed, in *Fyn*-deficient mice exposed to DSS to induce intestinal injury, the number of CD4+FOXP3+ cells was reduced as well as the capacity of lymphocytes to differentiate into regulatory T cells [40]. This indicates that Fyn has a protective role against intestinal injury. Using a similar approach it was demonstrated that Lyn also is protective against intestinal injury, microbiota-dependent intestinal inflammation and susceptibility to enteric pathogens [41]. Another mouse strain with an activated Lyn mutation (LynY508F) revealed a Lyn key role in the control of the innate immune response and validated its protective role against colitis [42]. 

### 3.3. SFKs in Intestinal Cell Transformation

Intestinal transformation is mainly caused by deregulated ISC proliferation [43], that can be mediated by SFKs (Figure 1C) [32]. Their transforming activity was revealed using genetically-modified animal models [32]. In CRC, the most frequent tumour-initiating event is abnormally elevated Wnt/beta-catenin signalling. Mechanistically, *Apc* inactivation leads to protein stabilization and activation of beta-catenin transcriptional activity [44]. Additionally, *Apc* inactivation leads to upregulation of SFK activity in the hyperproliferative “crypt progenitor cell-like” domain of the intestinal epithelium [32]. *Src* gene inactivation in IECs revealed its essential role in intestinal tumorigenesis-induced by *Apc* loss in mouse and fly models [32]. Specifically, conditional *Src* inactivation in IECs impairs adenoma tumour initiation and progression. However, this Src activity is not mediated by MAPK or STAT3 signalling, contrary to what is observed during intestinal regeneration [32]. In agreement, *Apc* inactivation also induces local inflammation due to a reduction in intestinal epithelial barrier function, which also contributes to tumour development [45]. Mechanistically, colon microbiota invasion activates IL-23-producing myeloid cells and expand tumour-resident IL-17-producing T lymphocytes, resulting in proliferation of transformed IECs through IL-11 receptor and gp130 upregulation, and eventually gp130/Src/YAP signalling [46,47]. Importantly, a positive autoregulatory loop controls gp130 and YAP expression, enabling tumour development. In agreement, in transgenic mice that express an active gp130 mutant in IECs, intestinal tumour development is accelerated upon *Apc* inactivation. Importantly, this gp130 pro-tumoral function depends on SFK activity [47]. The intestinal tumour-promoting role of other SFKs is largely unknown. Nevertheless, a recent study uncovered an important HCK function in colon tumorigenesis [48]. They observed that mutant mice carrying an oncogenic mutation in the *Hck* gene (HCKY520F) are prone to colon cancer development when treated with DSS and the carcinogen azoxymethane (AOM). This tumour-promoting effect was associated with HCK capacity to induce tumour-promoting M2-like macrophages and the accumulation of IL-6/lL-11 family cytokines [48].

## 4. SFKs in Human CRC

### 4.1. SFKs Deregulation

These animal studies suggest important SFK tumour-promoting activity in human CRC. Consistently, SFKs are frequently deregulated in CRC (80%) and their level of aberrant activity has been associated with poor prognosis [49,50,51]. Notably, SFKs are activated in intestinal polyps and adenoma [26,52] and SFK activity level is a prognostic factor of disease-free survival and overall survival in patients with early stage II-III CRC [53]. Active membrane localised FGR and HCK have been associated with prognosis and local inflammatory response, possibly by decreasing cytotoxic T lymphocytes in patients with stage I-III CRC [54]. Similar results were obtained in advanced CRC, suggesting additional SFK functions in cancer invasion. Notably, the level of Src and/or Yes tumour activity is a marker of bad prognosis and therapeutic resistance in stage IV CRC [49,55], while excessive Hck in tumour leukocytes was associated poorer CRC survival [48]. 

#### 4.1.1. SFK Upregulation

Despite this clinical evidence, SFK oncogenic roles were underestimated because SKK-encoding genes are rarely mutated in CRC [4]. Some oncogenic mutations for Src were previously reported in a small fraction of metastatic CRC [56], but these molecular alterations might not explain the frequent SFK deregulation observed in these cancers [49,50,51]. Several important mechanisms have been reported, that may explain how aberrant SFKs activity can promote CRC development (Figure 2A,B).

Src deregulation primarily involves gene upregulation and aberrant protein level [50]. For instance, Src is induced in early CRC, including in the subtype with elevated Wnt/beta-catenin activity (CSM2) [4,5,6,57]. Also, HIF1α-dependent SRC transcription is an important downstream event of hypoxia during CRC development [58]. Genomic studies identified *SRC* gene amplification in 5–10% of patients with CRC [4,7,57,59,60] suggesting that *SRC* oncogenic activation may implicate a similar mechanism as described for *HER2* [61]. Consistent with this, *SRC* gene copy number in stage IV CRC has been associated with left sided-tumours and liver metastases [59,60]. SFKs upregulation may also involve miRNA epigenetic mechanisms [50], such as mi-129-1-3p downregulation [62], and aberrant SFK protein accumulation, that can be induced by inactivation of the ubiquitination factor Casitas B-lineage lymphoma proto-oncogene (Cbl) or inhibition [63] of Src autophagic degradation through SNX10 inactivation [64].

#### 4.1.2. CSK Inactivation

However, Src upregulation is not sufficient to induce cell transformation because its activity is normally tightly regulated. Therefore, Src oncogenic induction requires additional kinase deregulatory mechanisms (Figure 2A,B). We propose that the combination of SFK upregulation and overactivation may explain, at least in part, the SFK’s essential role in CRC progression. Notably, one obvious mechanism of SFK deregulation relies on the inactivation of CSK, its main negative regulator. However, although *Csk* inactivation promotes SFK proliferative activity in animal models [29], this mechanism does not operate in human cancer, because *CSK* inactivation has been rarely detected in human CRC [4,16]. On the other hand, in human cancer, CSK activity can be prevented through a complex epigenetic mechanism via delocalization from the plasma membrane. Mechanistically, CSK cannot be recruited to the plasma membrane for effective SFK inhibition because of downregulation of the CSK-binding protein (Cbp), named transmembrane adaptor phosphoprotein associated with glycosphingolipid-enriched microdomains (PAG) [65,66]. Interestingly, PAG downregulation is mediated by promoter hypomethylation induced by oncogenic signals, including EGF, KRAS and Src [67]. However, functional studies suggest that PAG inactivation does not account for all Src transforming activities in CRC [66], possibly because of the contribution of additional Cbp (e.g., T-cell receptor adaptor proteins, Dok adaptors, SRCIN1) [68,69,70,71]. Besides, PAG can suppress Src signalling by additional CSK-independent mechanisms [65,72]. Surprisingly, CSK is upregulated in several CRC samples together with SFK activity, and anti-CSK autoantibodies have been detected in these patients, possibly representing a novel CRC biomarker [73]. These intriguing observations suggest that Csk could have some tumour promoting function in CRC. In agreement, the oncogenic pseudo-kinase and Src substrate Pragmin was identified as a novel Cbp [74] that can promote the CSK oncogenic role [75,76]. This novel function is mediated through Pragmin dimerisation that causes activation of CSK and phosphorylation of cell adhesive regulators, enabling epithelial cell dissemination [75,76,77]. 

#### 4.1.3. SLAP Inactivation

Recently, it was reported that Src tumour activity is under the control of the Src-like adaptor protein SLAP [78,79] (Figure 2A,B). Such negative regulatory mechanism mediated by small adaptor proteins was originally described for the Janus Kinase (JAK)/STAT pathway, which it is mediated by the suppressors of cytokine signalling (SOCS) [80]. SLAP comprises an N-terminal region, similar to the one in Src, and a unique C-terminus with binding affinity to CBL. *Slap*-deficient mice demonstrated the important role of this adaptor in lymphocyte development and activity, where it is strongly expressed [81]. SLAP is also abundantly expressed in intestinal epithelium and frequently downregulated in CRC (50%) [82]. Studies in CRC cell-lines and experimental CRC in mice revealed an important tumour-suppressor function for SLAP in CRC. SLAP controls Src tumour activity by promoting degradation of critical Src substrates, upon their aberrant phosphorylation, such as the adhesive receptor EPHA2 [82]. This SLAP activity implicates the association with the ubiquitination factor UBE4A [82], which is also involved in Crohn’s disease [83]. The nature of other critical Src substrates targeted by SLAP is currently unknown.

#### 4.1.4. SFK Post-Translational Modifications

Additional post-translational mechanisms were reported for Src activation in tumours, although their contribution to CRC development was not specifically examined. For instance, several protein tyrosine phosphatases (PTP), such as Receptor PTP alpha (RPTP alpha), PTP1B and PTP43A, have been implicated in the regulation of Src tumour activity, via kinase activation through Tyr530 dephosphorylation or kinase inhibition through Tyr419 dephosphorylation [50]. They could also affect Src activity by indirect mechanisms. For instance, SHP2 dephosphorylates PAG at Tyr317 to prevent CSK membrane activity [84] and RPTP alpha dephosphorylates the Src substrate paxillin at Tyr88 to control Akt signalling [85]. Finally, CREB binding protein-mediated Src acetylation induced by EGF or H2O2-mediated Src cysteine sulfenylation are novel post-translational mechanisms of Src activation [86,87]. The relevance of these novel regulatory mechanisms in CRC are unknown.

### 4.2. SFKs Signalling in Early CRC

#### 4.2.1. Wnt/Beta-Catenin and YAP Pathways

Although no Src transgenic mouse model recapitulates any of these oncogenic mechanisms, *Src*-deficient mice suggest an important Src function in the formation of Wnt/beta-catenin-dependent adenoma [32], in agreement with the preferential Src signature observed in CMS2 [5]. Consistently, important roles for Src and Yes activities were reported in the proliferation and tumour-initiating properties of CRC cells in which beta-catenin is active (also known as cancer stem cells, CSCs) [33,88]. However, Yes pathological function is not shared by Src, suggesting specific oncogenic activities among SFKs [88,89]. Mechanistically, SFKs can phosphorylate several regulatory components of this signalling cascade including beta-catenin to promote its transcriptional activity [50,90]. Recent reports identified YAP and tafazzin (TAZ) as additional mediators of SFK tumour activity in CRC [33,91]. This observation was corroborated by a strong correlation between phosphorylated YAP/TAZ and phosphorylated SFK levels in CRC [91]. Finally, SFK activity may allow the convergence between the YAP and beta-catenin pathways to maximise CRC development, whereby tyrosine-phosphorylated YAP can form a transcriptional complex with beta-catenin to induce CRC [33]. 

#### 4.2.2. RTK Pathway

Excessive SFK activity mediates the aberrant CRC cell response to growth/adhesive factors secreted by the tumour or its microenvironment (Figure 3) [50].

Notably, SFKs strongly influence RTK signalling at different levels. First, Src is an important effector of RTK signalling and a master controller of protein tyrosine phosphorylation in CRC cells, implicating the activation of many TK substrates [90,92,93,94]. Second, phosphoproteomic studies revealed a reverse signalling process where deregulated Src induces RTK activation [93]. Third, Src also phosphorylates several components of the MAPK and PI3K pathway, which are also effectors of RTKs signalling [90,93,94]. These Src-dependent mechanisms may perturb the CRC cells’ response to local extracellular cues and favour tumour progression. Additionally, Src phosphorylates many factors implicated in cell adhesion and morphology, including regulators of Rho signalling, consistent with its oncogenic role in CRC cell migration [90,93,94]. Surprisingly, these proteomic studies uncovered a large group of vesicular trafficking and mRNA maturation regulators, suggesting that deregulation of these molecular processes may also contribute to Src tumour activity [90,93,94]. Indeed, Src could facilitate CRC development by perturbing endocytosis or degradation of important growth/adhesive receptors [95]. Similarly, Src is a central trigger of cancer exosomes biogenesis by phosphorylating key components of the synthenin exosomal pathway [96,97]. These small extracellular vesicles are essential for reprograming recipient cells to facilitate tumour growth or angiogenesis [98].

#### 4.2.3. Cell-Cycle Progression

SFKs are also important regulators of cell-cycle progression [99,100,101,102], and excessive SFK activity may contribute to abnormal cancer cell division and chromosome instability in CRC, as reported in other transformed cells [103]. Notably, oncogenic Src induces cytokinesis failure, cell polyploidy and an excessive number of centrosomes [103]. Mechanistically, Src can promote delocalization of cytokinesis regulators including Aurora B and kinesin-like protein KIF23 and maintain YAP nuclear activity to weaken the tetraploidy checkpoint [103]. Src can induce mitotic slippage resulting in aneuploidy and therapeutic cell resistance by direct inhibition of the mitotic regulator Cyclin-Dependent Kinase 1 [104]. These Src mitotic defects may also contribute to CRC cell invasive behaviour. For instance, active Src hijacks mitosis to extrude transformed cells from the epithelium, a process involved in early cell dissemination [105]. Similarly, centrosome amplification leads to increased Rac1 activity that disrupts normal cell-cell adhesion and promotes invasion [106].

### 4.3. SFKs Signalling in Advanced CRC

#### 4.3.1. CRC Angiogenesis, Survival and Metabolism

Studies in human CRC cells and in CRC samples indicate an important SFK role also in CRC progression (Figure 3). In advanced CRC, wild-type Src expression is highly oncogenic due to defects in kinase regulation and substrate degradation, and promotes tumour growth and liver metastasis in nude mice [93]. Active SFKs may influence several important steps of CRC metastasis development. First, Src plays an essential role in the induction of tumour angiogenesis during CRC progression [94]. A complex mechanism may be involved, including *SRC* upregulation by hypoxia [58], induction of angiogenesis factors [107] and Src-mediated angiogenic receptor signalling [108]. Second, excessive SFK activity sustains tumour cell survival during CRC progression [109]. This function is mediated by activation of Akt survival signalling [109], inhibition of FAS apoptotic signalling [110,111], and induction of antiapoptotic genes, such as BCL2L1 or BIRC5 [33]. Third, Src can respond to the strong tumour energy demand by phosphorylating and activating several metabolic cascades. For instance, Src can promote intestinal tumour development by direct phosphorylation of the glycolytic enzyme PFKFBP3 enabling aberrant anabolic glycolysis [112]. Src may deregulate mTORC1 activity, a master regulator of protein synthesis, by overriding its inhibition by Gator1 [113]. 

#### 4.3.2. CRC Cell Dissemination and Colonization

SFK activity is a key promoter of CRC cell dissemination. Active SFKs induce a migratory state that resembles the epithelial to mesenchyme transition phenotype observed during embryogenesis [114]. Specifically, SFKs induce disruption of cell junctions by phosphorylating components of the E-cadherin junctional complex, thus enabling beta-catenin induction of migratory genes [115]. Fyn mediates CRC cell migration induced by the noncanonical Wnt pathway through the Frizzled 2 receptor, specifically in high-grade CRC that strongly expresses Wnt5a/b ligands [116]. Src may also induce CRC cell invasion by producing basal actin-enriched adhesive structures called invadosomes from which metalloproteases are secreted for extracellular matrix degradation [117]. Metastases originate from distant tissue colonization by a small population of disseminated cells with CSC activity that require high beta-catenin activity [118]. Although not formally established in CRC, active Src may participate in this process by supporting beta-catenin signalling and by modulating the morphology of circulating tumour cells [119,120]. Similarly, Src and Lyn activities mediate CSC activity and migration induced by the hyaluronic receptor CD44 and its ligand osteopontin [121,122,123]. Src and Yes activity may also promote endothelial permeability for effective CRC cells colonization [124]. Metastatic induction requires formation of the so-called premetastatic niche that supports the landing of disseminated tumour cells and their interaction with the host tissue [3]. This process originates from the primary tumour or from other metastases and is mediated by long-range communication implicating cancer exosomes filled with metastatic signals [98]. Therefore, Src could facilitate metastatic niche formation by promoting tumour exosome production [96,97]. Additionally, specific adhesive components deposited at the metastatic site (e.g., tenascin C, periostin, collagen) may facilitate CRC cell adhesion, survival and CSC activity through a Src-dependent mechanism [125]. 

#### 4.3.3. Metastasis Development

Intriguingly, CRC metastasis seeding can be an early event of CRC for a large fraction of patients [126,127] suggesting that these dormant cells must be reactivated by specific signals during CRC progression [3]. It was recently reported that Src is an important mediator of this malignant process in other epithelial tumours where it is activated by specific adhesive components and cytokines [128,129,130]. This raises the attractive idea that Src deregulation in dormant CRC cells could facilitate metastatic reactivation. Experimental evidence also supports an important SFK function in metastatic growth because inactivation of Yes or Src activity reduces CRC liver metastasis in nude mice [131], while SLAP silencing enhances metastatic development [82]. Finally, some evidence suggests that aberrant SFKs could favour tumour immune evasion. For instance, the Srci dasatinib enhanced immune infiltration and tumour response to anti-Programmed Death-ligand 1 (PD-L1) antibodies in experimental solid tumours, including CRC [132,133]. Results in other tumour types suggest a role of active Src in the production of cancer exosomes to modulate cancer immunity [98], and in the expression of the immune suppressive protein PD-L1 induced by oncogenic signals [134].

## 5. Therapeutic Strategies to Target SFKs Signalling in CRC

### 5.1. Therapeutic Utility

Accumulated evidences obtained in experimental CRC models suggest that SFKs could be attractive targets in advanced CRC. SFK inhibition may be of clinical value in advanced CRC due to their role in CRC cell dissemination and CSC activity, which are the main causes of tumour relapse and metastatic progression. In agreement, specific Srci reduce liver metastasis development in nude mouse models, an effect associated with decreased tumour angiogenesis, cell proliferation and survival [131,135]. However, it was not assessed whether SFK inhibition reduces CSC activity. In addition, Src oncogenic role in RTK signalling may explain why Srci sensitises CRC to RTK inhibitors in experimental CRC models [136]. Similarly, the effect of Src activity on MAPK/PI3K signalling is consistent with findings showing the potential clinical utility of combining Srci with KRAS effector inhibitors (MAPK kinase, and PI3K inhibitors) in KRAS mutant CRC [137,138]. This observation is clinically relevant because these tumours are refractory to the upstream EGFR antibody currently used in the clinic, cetuximab [11]. Src activity has been identified as a mechanism of tumour resistance to oxaliplatin in metastatic CRC, suggesting that its pharmacological inhibition could enhance the efficacy of oxaliplatin-based chemotherapy in patients with CRC [55,139]. Finally, some evidences suggest that Srci might sensitise CRC to anti PD-L1 immune checkpoint inhibitors [132,133]. 

### 5.2. Therapeutic Strategies

Several Src-like ATP competitive inhibitors have been developed for oncology, including dasatinib, bosutinib and saracatinib [12]. Although dasatinib and bosutinib were originally developed to target Src/Abl activities, they are multikinase inhibitors [140,141,142]. Moreover, tirbanibulin, a Src-like peptide binding site inhibitor, inhibits also tubulin polymerization [143]. Although most of these Srci display significant anti-tumour activity in experiment tumour models, they gave disappointing results in patients with CRC, both as monotherapy and in combination with the current therapies [144,145,146]. For instance, the combination of dasatinib with the chemotherapy regimen FOLFOX (folinic acid, 5-fluorouracil, and oxaliplatin) with or without cetuximab did not show any meaningful clinical activity in refractory CRC [146]. 

The complex mechanisms of Src regulation and hyperactivation in CRC discussed above may explain the lack of anticancer effect of these drugs, particularly the lack of patient stratification, drug efficacy and selectivity, resulting in significant toxicity. These complex mechanisms also suggest new therapeutic strategies to better target Src signalling in CRC (Figure 4).

For instance, several noncatalytic strategies could be developed to improve Src inhibition in CRC, including allosteric inhibitors of the myristoyl switch regulatory mechanism, as recently demonstrated with asciminib in BCR-ABL expressing chronic myeloid leukaemia [147]. Small molecules that interfere with Src UD signalling, kinase dimerisation or membrane localisation by disrupting UNC119-Src interaction may lead to Srci with higher specificity and lower toxicity [148]. Additionally, drugs that reactivate SRC inhibitors, such as PAG or SLAP, could limit CRC invasion or metastatic reactivation. Finally, patient selection based on Src activity level in CRC would clearly improve the overall therapeutic response. While CMS2 tumours should preferentially respond to Srci, studies in advanced CRC suggest that left-sided tumours with elevated RTK signalling are also good targets [7,93]. Moreover, there is no validated biomarker for Src-dependent tumours. Nevertheless, several candidates could be proposed, such as high Src tumour activity or *SRC* copy number, high phosphorylation of Src effectors, such as FAK, YAP/TAZ, RTKs, or even high tyrosine phosphorylation level. However, high SLAP expression could limit Src oncogenic signalling, and therefore, tumours could be less responsive to Srci, despite their high aberrant Src activity [82]. Therefore, SLAP expression could be an additional predictor of the tumour response to Srci.

## 6. Conclusions and Perspectives

Since the first observation of abnormal Src activity in CRC samples [149], much has been learned about SFK physiological and oncogenic functions in the intestine. Although, their oncogenic roles have been underestimated because of the absence of frequent somatic mutations in CRC, there is now strong evidence of their detrimental role in CRC cell invasion. This suggests that Srci could be useful for the management of patients with metastatic CRC. However, Srci clinical utility in CRC has not been demonstrated yet, because of lack of patient stratification, drug efficacy and selectivity. Clearly much more needs to be learned about how SFKs function during CRC development to reach this objective. Recent molecular studies have highlighted the much higher complexity of SFK regulation, which needs to be investigated in order to efficiently target these activities. Future studies on SFK physiological roles in the intestine may bring important insights into SFK influence on CRC development. Moreover, appropriate CRC models are crucially needed, including genetically modified mice, to recapitulate some of the activating mechanisms reported in human CRC, in order to assess the complexity of Src signalling. It would be important to analyse the respective oncogenic roles of these SFKs in the epithelial and microenvironment compartments of these tumours. Moreover, their contribution to metastatic reactivation and immune evasion also are important questions that could be addressed with these models. Finally, phosphoproteomic studies are needed to decipher the molecular complexity of Src signalling in CRC. For instance, the large group of mRNA regulators identified in such studies [90] points to an unsuspected feature of Src tumour activity. Overall, future studies should allow understanding of how SFKs regulate epithelial homeostasis and tumorigenesis, and improving Src-based therapies in CRC.

## Figures and Tables

**Figure 1 cancers-12-02014-f001:**
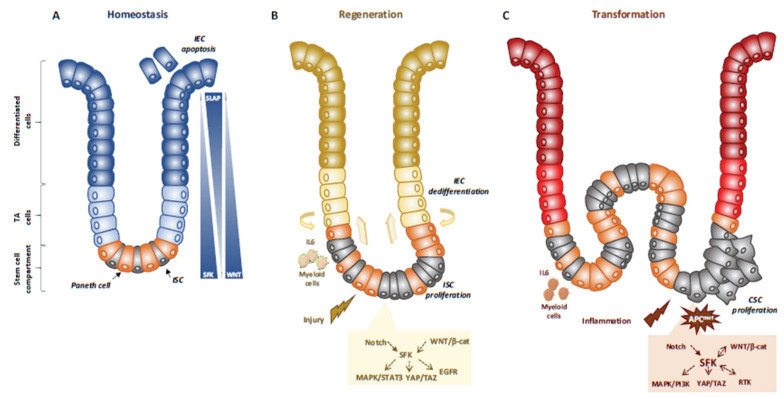
SFKs in intestinal homeostasis, regeneration and transformation. (**A**) SFKs regulate ISC proliferation, Paneth cell differentiation and IEC survival during intestinal homeostasis. SFKs, Wnt and Slap activity in the intestinal epithelium is indicated. (**B**) SFKs mediate intestinal regeneration by activating ISC proliferation. (**C**) SFKs mediate tumour formation induced by activating CSC survival. Src signalling involved in intestinal regeneration and transformation is indicated.

**Figure 2 cancers-12-02014-f002:**
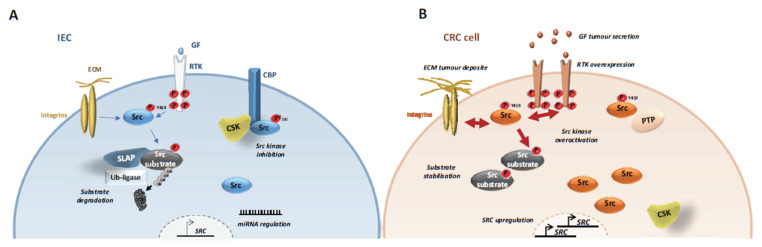
Mechanisms underlying SFK tumour activity in CRC cells. SFK tumour activation implicates abnormal *SRC* gene upregulation via *SRC* amplification, transcription and protein stabilisation. Additionally, SFK kinase activity is deregulated via downregulation of its negative regulator CSK and overactivation of their upstream receptors. Finally, SFK signalling is elevated by abnormal substrate stabilization via SLAP inactivation (**A**) Src signaling in normal intestinal epithelial cells (IEC). (**B**) Src signalling in CRC cells.

**Figure 3 cancers-12-02014-f003:**
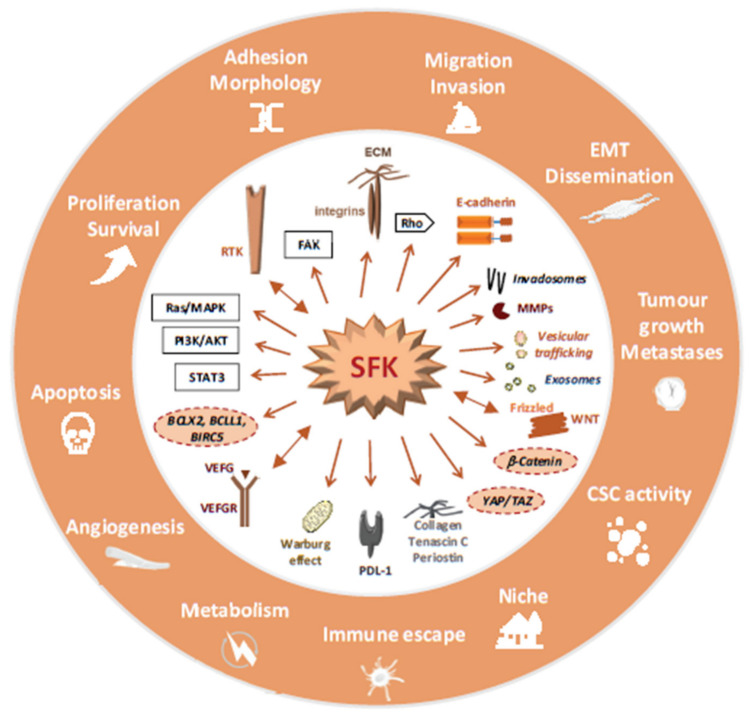
A model summarizing SFK signalling in CRC, molecular and cellular processes activated by excessive SFK activity in CRC is highlighted.

**Figure 4 cancers-12-02014-f004:**
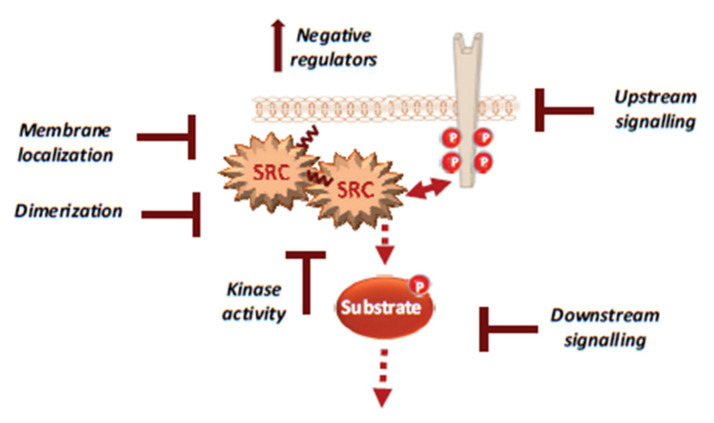
Therapeutic strategies to efficiently target Src signalling in CRC, including drug inhibition of Src UD signalling, kinase dimerisation, membrane localisation, drugs that reactivate Src inhibitors and drugs that inhibit activity of downstream signals.

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
