# Peer review of "Src Family Tyrosine Kinases in Intestinal Homeostasis, Regeneration and Tumorigenesis"

_cancers, 2020, doi:10.3390/cancers12082014_

Round 1

Reviewer 1 Report

Because SFK-encoding genes are not frequently mutated in human cancers, the relationship between SFK and tumorigenesis of intestine are not clear. Due to these reasons, this review is not easily readable. Research results regarding SFK in interstinal tumors are listed, but it is not easily understandable.
I suggest to reduce the contents regarding tumorigenesis, and focus on the homeostasis and regeneration. Explanation of novel regulatory mechanisms of SFK will be good.

Page 2, line 15: MAPK and PI3K are not tyrosine kinases (TK)
Page 7: SKF -> SFK

Author Response

We thank the reviewer for their comment.

The reviewer questions the relationship between SFK function in intestinal homeostasis and tumorigenesis. However, we feel that linking the role for SFK in intestinal regeneration with tumorigenesis gives some important hints into how SFK may work during early tumorigenesis. For instance, the SFK/YAP axis found in CRC was originally characterized from intestinal regeneration.

The reviewer also raises the complexity of signalling proteins, whose oncogenic activity is induced by non-genetic mechanisms. We feel this is an important point to discuss because several drugs targeting such molecules are currently used in the clinic. Because of moderate activity in some patients, a better understanding on how these proteins contribute to tumorigenesis is key to improve targeted therapies. One seminal example comes from the clinical activity of drugs targeting the RTK EGFR in CRC, although this RTK is not frequently mutated in CRC.

The reviewer was concerned about the clarity on the part focused on SFK signalling in intestinal tumours. It was however noticed that this lack of clarity was not raised by the other reviewer. However to address the reviewer's concern and improve the overall clarity of our ms, we restructured this part with clear sub-paragraphs on SFK signalling in early CRC (4.2.1 Wnt pathway; 4.2.2 RTK pathway; and 4.2.3 cell-cycle progression); and in advanced CRC (4.3.1. tumour angiogenesis, survival and metabolism; 4.3.2 CRC cell dissemination; and metastasis development 4.3.3). This part is supported by a schematic described in Figure 3.

p12: MAPK and PI3K are downstream components of TK signaling and this pint has been clearly stated in the reviewed ms.

SKF has been changed to SFK thoughout the ms.

Reviewer 2 Report

In the manuscript, the authors have reviewed recent findings on SFK signaling during intestinal homeostasis, regeneration, and tumorigenesis. The manuscript is a continuation of authors' previous work (Sirvent A, Benistant C, Roche S. Oncogenic signaling by tyrosine kinases of the SRC family in advanced colorectal cancer. Am J Cancer Res. 2012;2(4):357-371).

My comments:

  1. The authors use the phrase "recent" to cite works from 1989 (item 149 in References), i.e. 31 years ago.
  2. Polyps are considered a precancerous stage. Although not all colonic polyps are adenomas and more than 90% of adenomas do not progress to cancer. For this reason, it would be worth presenting changes in Src activity in colorectal polyps.
  3. Currently, some Src inhibitors, such as Dasatinib, are registered as anti-leukemic drugs. Are drugs with the same mechanism of action allowed for the treatment of colorectal cancer?
  4. The quality of English language has to be improved.

Author Response

We thanks the reviewer for their comments.

Regarding their specific comments:

  1. The word "recent"  (p16) is not present in the revised ms.
  2. Src activation in polyps has been now specified in the revised ms (p7 §4.1)
  3. Currently, there is no anti-leukemic drugs with a similar type of mechanism to dasatinib that is allowed for the treatment of colorectal cancer. However we could recently show that the anti-leukemic drug nilotinib displays anti-metastatic activity in experimental CRC models through the inhibition of the collagen receptor DDR1 (Jeitany EMBO Mol Med 2018). This suggests that nilotinib could be of interest for the treatment of advanced CRC.
  4. Our ms was checked by a native english speaking professional and consistently, both reviewers indicated that the used english was correct and readable. However, our revised ms was double-checked by a native english scientist and additional errors have been corrected.

Reviewer 3 Report

In this review entitled "Src family tyrosine kinases in intestinal homeostasis, regeneration and tumorigenesis", Sirvent et al reviewed the role of SFK signalling during intestinal homeostasis, regeneration and tumorigenesis. The authors also highlighted mechanisms that could be exploited in therapeutic strategies to target SFK signalling in colorectal cancer.

This work needs to consider (and revise):

  1. The points added in the attached file.
  2. Figures are not at its best quality.
  3. Further revision required to confirm the usage of the correct  abbreviations and it is consistent (Src vs. SRC).

Author Response

In this review entitled "Src family tyrosine kinases in intestinal homeostasis, regeneration and tumorigenesis", Sirvent et al reviewed the role of SFK signalling during intestinal homeostasis, regeneration and tumorigenesis. The authors also highlighted mechanisms that could be exploited in therapeutic strategies to target SFK signalling in colorectal cancer.

We thanks to reviewer for their comments and suggestions.

This work needs to consider (and revise):

  1. The points added in the attached file.The points added in the attached file have been addressed in the revised version of our manuscript.
  2. Figures are not at its best quality.A new set of figures of better quality is now been provided.
  3. Further revision required to confirm the usage of the correct  abbreviations and it is consistent (Src vs. SRC).The usage of abbreviation has been checked throughout the ms and the figures and modified according to the reviewer's suggestion (Src refeers to the protein, Src to the mouse gene and SRC to the human gene).

Round 2

Reviewer 1 Report

The authors did not revise the manuscript according the the comments.
I suggest to revise the manuscript, according to the comments provided at the first review.

Author Response

I suggest to revise the manuscript, according to the comments provided at the first review.

The reviewer mentioned that because SFK-encoding genes are not frequently mutated in human cancers, the relationships between SFK and tumorigenesis of intestine are not clear and due to these reasons, this review was not easily readable. He/She suggested to reduce the contents regarding tumorigenesis, and focus on the homeostasis and regeneration. To be honest, we were confused and did not quite understand their main concern. Our review is dedicated to the special issue of Cancers on SFK in human cancer, and one of our main objective was to discuss on the importance of linking intestinal regeneration and tumorigenesis because it gives important hints on how SFK may work in early tumorigenesis. Several oncogenes display tumour activity in the absence of somatic mutation and we feel it is important to understand how such oncogenic molecule can be activated by non-genetic mechanisms to promote cancer formation. Such study may bring important insight into the complexity of tumor development and improve therapeutic strategies aiming at blocking their tumor activity. One seminal example is the clinical utility of targeting EGFR in advanced CRC, which is however not mutated in these cancers; it is now clear that a better understanding on how EGFR promotes metastasis leads to therapeutic strategies aiming at overcoming therapeutic resistance observed in a large fraction of cancer patients. However, in order to address their concern on a lack of clarity, we had restructured the part on SFK signaling in CRC in our revised ms in order to address this issue.